# Direct and moderating effects of environmental regulation intensity on enterprise technological innovation: The case of China

Xiguang Cao[1], Min Deng[1], Fei Song[2], Shihu Zhong[3]*, Junhao Zhu[1]

**1** School of Urban and Regional Science, Institute of Finance and Economics Research, Shanghai University of Finance and Economics, Shanghai, China, **2** School of Information Engineering, Jiangsu Open University, Nanjing, China, **3** Department of Investment, School of Public Economics and Administration, Shanghai University of Finance and Economics, Shanghai, China

☯ These authors contributed equally to this work.
\* zhongshihu@163.sufe.edu.cn

**Data Availability Statement:** The data underlying the results presented in the study are available from China Statistical Yearbook 2009-2017, China Industry Statistical Yearbook 2009-2017, China

## Abstract

There is few significant attempt to integrate environmental regulation, government financial support, and corporate technological innovation in a methodological framework. Employing the data of the industrial enterprises with an annual turnover of over 20 million yuan from 30 Chinese provinces or municipalities between 2008 and 2016, this paper applies the fixed effect regression model to reveal the relationships between environmental regulation, government financial support, and corporate technological innovation simultaneously. Results show that: (1) there exists a U-shaped relation between environmental regulation intensity and technological innovation of enterprises which declines first and then climbs up, and China is still at the stage of inhibition before the "inflection point". (2) government financial support does not significantly work on technological innovation directly, but environmental regulation drives this effect to be achieved; when the value of ln$ER$ is higher than 3.69, government financial support can significantly facilitate corporate technological innovation. (3) the comparison between regional samples reveals that heterogeneity exists in the influence of environmental regulation intensity and government financial support on corporate technological innovation. The threshold value of enabling effects of environmental regulation in eastern region is higher than that of the central and western region. These results remain consistent after we experiment several robustness checks. Theory and policy implications of our work are discussed.

## Introduction

China's economy sustained high-speed growth as industrialization and urbanization in the country advanced by leapfrog. However, pollutant emission increased and environment quality was sacrificed. According to *Chinese Economic-Ecological Product Accounting Research*

Statistical Yearbook on Science and Technology 2009-2017, China Energy Statistical Yearbook 2009-2017, and Wind Information Database.

**Funding:** We received no external funding during this specific study, and the funders had no role in study design, data collection and analysis, decision to publish, or preparation of the manuscript.

**Competing interests:** The authors have declared that no competing interests exist.

*Progress Report 2018*, the accounting results of gross economic-ecological product (also known as "green GDP") jointly released by the Ministry of Ecology and Environment of the People's Republic of China and the National Bureau of Statistics of China, ecological and environmental costs of 31 Chinese provinces in 2015 amounted to two trillion *yuan*, increased by 106.2% compared with 970.11 billion *yuan* in 2009. This demonstrates growing environmental costs in China's economic development. Therefore, China has to upscale its environmental regulation intensity before the country reaches the upper limits of its ecological environment. However, poverty alleviation in China remains arduous and social welfare is still at a relatively low level, which means the country has to consider, when implementing environmental regulation, the economic implications entailed by them [1–4]. Technological innovation is the decisive factor in realizing "win-win" goals in both environmental protection and economic development [5–7]. The innovation-enabling effects of environmental regulation should become the focus of research. Is it a positive "offset effect" or a negative "counterbalance effect"? Researchers draw varied conclusions based on different premises, ways of analysis, and research samples.

On the one hand, the traditional *neoclassical camp* proceeds from a static perspective and argues that the dilemma between environmental regulation and technological innovation is implacable. Because under the premise that technology, resource allocation, and consumer demand were fixed, the introduction of environmental regulation would increase the cost of an enterprise, lowering its innovation capacity and competitiveness in international market [8–9]. On the other hand, some researchers adopt a dynamic angle and prove the possibility of a win-win scenario between environmental regulation and improvements on technological innovation [10–11]. They point out that properly crafted environmental regulation can induce, under dynamic constrains, enterprises to improve resource allocation efficiency and technology and trigger "innovation offsets" so as to partially or even fully offset the costs of complying with them. These studies fails to offer a consistent explanation on whether the effect of environmental regulation on enterprise technological innovation is positive or negative. But the fact that environmental regulation does affect enterprise technological innovation is agreed, which serves as an enlightenment for this paper [12–13].

Moreover, government financial support for corporate technological innovation such as direct funding (R&D subsidies through fiscal appropriation) and tax deduction and exemption after R&D spending are both important sources of innovation fund for enterprises and exterior incentives for them to innovate. Enterprise technological innovation is also a process of interaction among several elements [14–15]. In the case of limited government financial support, the effect of environmental regulation on corporate technological innovation and the extent to which enterprise technological innovation is influenced by them may evolve, through which environmental regulation's moderating impact on corporate technological innovation is uncovered.

To summarize, although there are some studies on the factors impacting the achievement of technological innovation, they are limited in discussing one specific factor, such as environmental regulation or government financial support; while there is few significant attempt to integrate environmental regulation, government financial support, and corporate technological innovation in a methodological framework. Therefore, the industrial enterprises with an annual turnover of over 20 million yuan from 30 Chinese provinces or municipalities (Tibet is excluded as its data are not complete) between 2008 and 2016 were adopted in this paper to conduct the empirical study to contribute to extant literature in the following three aspects: first, government financial support is incorporated into the model as an important factor in the choice of the corporate technological innovation strategy, and thus the direct and moderating effects of environmental regulation intensity on enterprise technological innovation is

examined simultaneously. Second, based on the measurement of environmental regulation intensity and government financial support level, the authors employ the panel data analysis model to explore whether an "inflection point" exists in promoting enterprise technological innovation through environmental regulation and government financial support. Third, the authors examine whether a regional heterogeneity exists due to regional differences in environmental regulation intensity and government financial support level.

## Literature review

Our article relates to two main strands of the literature. The first category of literature concentrates on environmental regulation's effects on enterprise technological innovation, and it was first mentioned by Magat (1978) who believes that innovation is the key to dissolve the contradiction between environmental protection and economic development [16]. But opinions on the relationship between these two diverge. On the one hand, traditional neoclassical theory holds that environmental regulation can improve social benefits on the whole but will inevitably increase the costs of enterprises, lowering their capability in technological innovation. This constitutes the "compliance costs" entailed by environmental regulation. Gray (1987) reveals, based on the data from American manufacturing and power industry, the negative effects of environmental regulation on innovation level and economic growth as environmental regulation would increase enterprises' costs in emission reduction and lower innovation inputs [8]. Employing different samples, Palmer et al. (1995), Brännlund et al. (1995), and Barbers and McConnell (1990) prove that environmental regulation crowds out firm-financed spending on technological innovation and conclude that stringent environmental regulation would drive enterprises into a deteriorating situation [17–19]. These scholars believe that environmental regulation is, under a static premise that technologies of enterprises, production procedure, and consumption demand remain unchanged, detrimental to technological innovation in enterprises.

On the other hand, some scholars adopt a dynamic perspective and explore environmental regulation's compensation effects on corporate technological innovation in a bid to construe a win-win model between environmental protection and economic growth. Porter and van der Linde (1995) argue that "innovation offsets" brought by technological progress overpower environmental regulation costs as properly crafted environmental regulation can effectively induce regulated enterprises to innovate, hence the coinage of *Porter hypothesis* [11]. Jaffe & Palmer (1997), Berman & Bui (2001), and Hamamoto (2006) examine the relation between environmental regulation and enterprise technological innovation based on samples from the U.S. and Japan, the results of which show the pressure put by environmental regulation can galvanize technological innovation among enterprises, testifying *Porter hypothesis* [20–22]. Acemoglu et al. (2012) further divide production department into "clean" ones and "dirty" ones [5]. Through construing a technological progress model, they systematically deduct the endogenous process of technological advancement and analyze impacts of environmental regulation on corporate technological innovation. Their numerical simulation results show that the combination of pollution tax and government financial support can facilitate innovation in clean technology and reduce emission while sustaining economic growth. In addition, some researchers believe that there exists a nonlinear relation between environmental regulation intensity and enterprise technological innovation, which presents a U-shape, namely a threshold value exists for environmental regulation to deliver effects [23–24].

The second category of literature focuses on the impacts of government financial support on corporate technological innovation. Indeed, technological innovation is an investment which can benefit enterprises. However, risks of innovation, constrains on external financing,

and overflow of innovation hinder enterprises from innovating. Government usually adopts measures like project funding, tax reduction or exemption, or R&D subsidy to motivate and support technological innovation in enterprises [25–26]. Currently, the research results of government financial support's impact on technological innovation by varied scholars are contentious. Researchers such as Clausen (2009), Bronzini & Piselli (2016), and Czarnitzki and Lopes-Bento (2014) hold that government financial support can lower enterprises' R&D costs and risks, thus promoting their inputs in technological innovation [27–29]. Of this, Bronzini & Piselli (2016) evaluate the impact of government financial support plan on patent applications in northern Italy through regression discontinuity and find that the subsidy plan boosts the possibility of patent applications significantly [28]. In addition, Broekel et al. (2017) claim that government financial support impact region's innovation growth when providing access to related variety and embedding regions into central positions in cross-regional knowledge network [30]. Other researchers like Wallsten (2000), Görg & Strobl (2007) reveal that government financial support crowds out firm-financed R&D spending [31–32].

Based on the above research status, we find that it is generally accepted that environmental regulation and government financial support do affect enterprise technological innovation, whereas there is no reasonable and unified explanation of the influence directions and degrees of environmental regulation and government financial support on corporate technological innovation [33–35]. Moreover, most studies focused on discussing the relationship between one specific factor (environmental regulation or government financial support) and enterprise technological innovation. However, there are few significant attempt to integrate environmental regulation, government financial support, and corporate technological innovation in a methodological framework [36–39]. Zhao and Sun (2015) apply the panel regression models to investigate the effect of environmental regulation on corporation innovation and competitiveness using Chinese pollution-intensive corporation panel data for 2007 to 2012 [38]. Guo, Qu & Tseng (2017) use empirical data on 30 provincial administrative regions in China during 2011 to 2012 by employing structural equation modeling (SEM) approach to explore the effect of environmental regulation on technological innovation and green growth performance [39]. These existing studies do lay the foundation and offer some inspiration for this paper, however, they emphasis on the direct effect of environmental regulation on enterprise technological innovation, ignoring exploring the moderating effects of environmental regulation on enterprise technological innovation at the regional level, which resulting from the interaction effects of environmental regulation and government financial support on corporate technological innovation. Thus, this paper develops an integrated model to investigate the relationship of environmental regulation, government financial support, and corporate technological innovation. Which aims to contribute to the nascent literature in technological innovation practices by incorporating government financial support into the model as an important factor and investigating the direct and moderating effects of environmental regulation intensity on enterprise technological innovation simultaneously.

## The status quo analysis on environmental regulation, government financial support, and enterprise technological innovation

The Chinese government takes diverse measures on environmental regulation. They can be grouped into three categories: administrative measures, incentive-based measures, and willingness-based ones. As early as 1980s, policies represented by emission restriction on industrial waste, environmental impact report system, and emission permits were piloted in designated places and then applied to the whole country. Administrative measures focus on both the control of traditional sources of pollution and the prevention of new sources of

pollution. The regulated area was also expanded from specific region to the entire nation. In addition, administrative measures were applied through stipulating various environmental standards, quota standards, or issuing bans. They are authoritative and can be employed rapidly, positioning themselves as the major forms of environmental regulations in the early years.

Since 1990s, incentives like discharging fees, subsidies, emission rights trading system, cash deposit for pollution control operation, and comprehensive preferential policies on tax have been adopted in various provinces first and replicated nationwide later. The purpose is to induce enterprises to invest in environmental protection, offer them with the freedom to choose, enhance their engagements in environmental protection so as to secure green production. On the one hand, the government urged enterprises through environment tax and fees and deposit refund. On the other hand, the government granted financial benefits through environmental protection subsidy and emission rights trading system to ensure that enterprises can gain from pollution control.

Starting from the 21ˢᵗ century, willingness-based measures such as information disclosure, public engagement in supervision, environmental qualification, and environmental agreement have been in wide use. In the case of administrative orders or incentive-based regulation, the autonomy of enterprises is hampered, driving them to invest in environmental protection and stunt technological innovation. Meanwhile, enterprises shoulder relatively low costs of environmental fine and environmental protection in order to improve environmental performance. The return on environmental protection investment is, therefore, relatively high. Considering this, willingness-based environmental regulation can enhance the environmental awareness of enterprises and offer a higher degree of autonomy in technological innovation.

As the ecological environment in China continues to deteriorate, treatment cost per unit of pollutants rises. Fig 1 showcases treatment costs for major pollutants in China from 2008 to 2016. Currently, solid waste, waste gas, and liquid waste are major pollutants in China. The figure indicates that the overall costs for waste treatment are on the rise from 2008 to 2016. Of this, investments in waste gas treatment are the highest, followed by those in liquid waste.

Government financial support for R&D activities in China are granted through direct R&D subsidies or indirect ways such as tax refund and exemption. Referring to the practices of Guellec & Van Pottelsberghe de la Potterie (1997) and Guo et al. (2016), the article just selects the authors adopt the government-financed fund in R&D expense in the total R&D expense of industrial enterprises with an annual turnover of over twenty million *yuan* to evaluate government financial support for R&D activities. As shown in Fig 2, government financial support for R&D activities witnessed a steady growth from 2008 to 2016. It has increased from 13.658 billion *yuan* in 2008 to 41.91024 billion *yuan* in 2015, up by over two-fold. But, the year of 2016 experienced a slight decline.

Fig 3 presents a horizontal comparison on government financial support for R&D activities among 30 Chinese provinces or municipalities. The authors compare and contrast the average government R&D subsidy of different provinces or municipalities from 2008 to 2016. As indicated in Fig 3, there is a huge difference in government financial support for R&D activities among varied regions. Manifestly, government financial support for R&D activities in eastern region such as Guangdong Province, Shandong Province, Shanghai Municipality, Jiangsu Province, and Beijing Municipality is high. Government financial support for R&D activities is low in western region except for Shaanxi Province. Government financial support for R&D activities in central China is greater than those of the eastern and western region.

China is in the transition from an extensive economy to an intensive economy. Enterprise technological innovation is the key to and plays a pivotal role in overcoming the bottleneck during economic development and dissolving the contradiction between economic

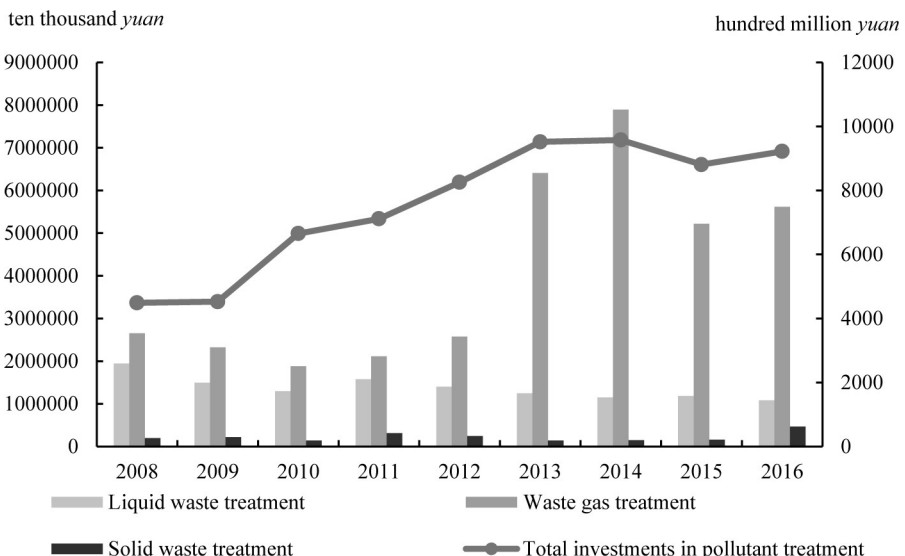

**Fig 1. Investments in pollution abatement from 2008 to 2016 in China.**

development and environmental protection. Enterprise R&D expense and successful patent applications are significant indicators for corporate technological innovation. As demonstrated in Fig 4, enterprise R&D expense in 2008 was 286.5 billion *yuan* and it registered a steady growth toward 1.0945 trillion *yuan* in 2016, increased by a factor of four within seven years, representing an annual growth rate of 21%. Successful patent applications in 2008 were 173.6 thousand and experienced a continuous increase toward 715.4 thousand in 2016, up by over four times within seven years, showcasing an annual growth rate of 22%.

The authors compare corporate R&D expense and successful patent applications between 30 Chinese provinces or municipalities to shed light on the difference of enterprise technological innovation level in different regions. As indicated in Fig 5, enterprise R&D expense in varied regions comes in line with successful patent applications. For instance, corporate R&D

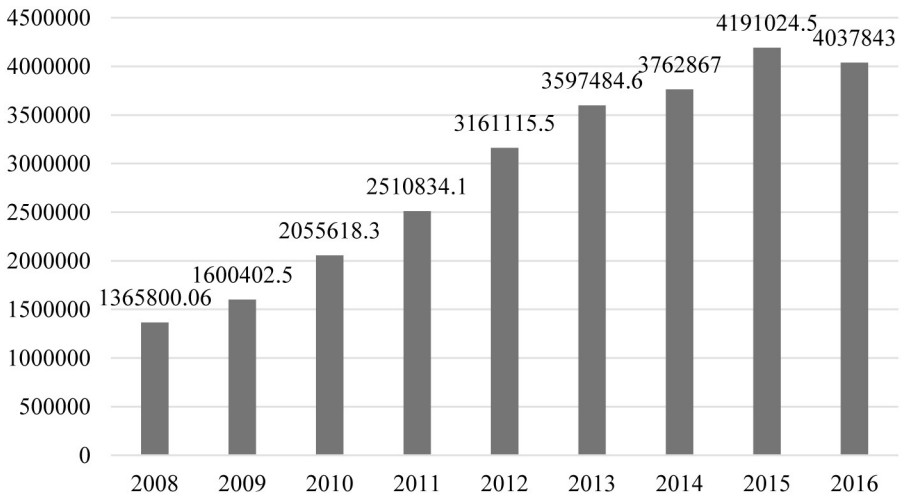

**Fig 2. Evolving trend on government financial support for R&D activities from 2008 to 2016 in China (ten thousand *yuan*).**

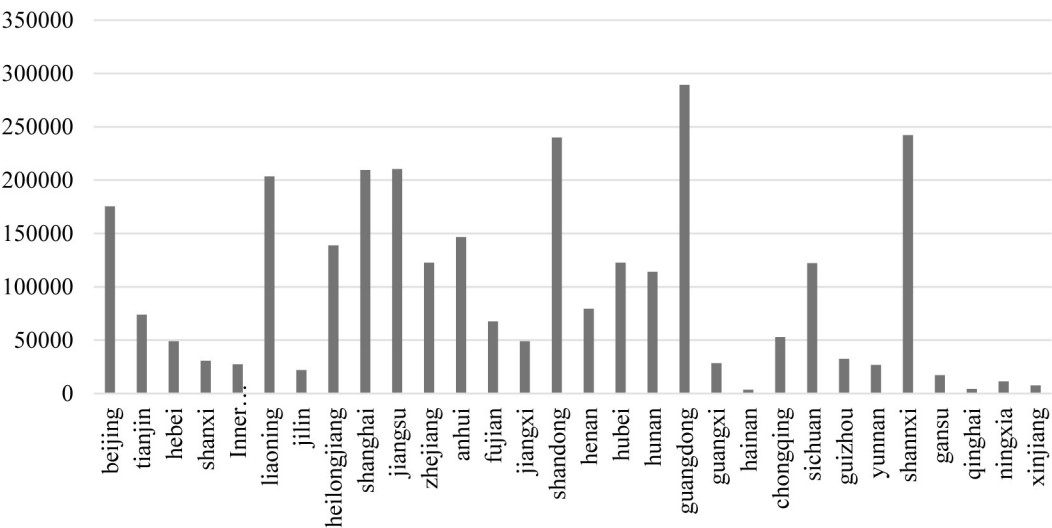

**Fig 3. Comparison on Government financial support for R&D activities among Chinese provinces or municipalities (ten thousand *yuan*).**

expense in provinces such as Guangdong, Jiangsu, Zhejiang, and Shandong are higher than other provinces or municipalities. Successful patent applications are accordingly higher in the abovementioned provinces, which are all located in eastern region. Comparatively, enterprise R&D expense and successful patent applications are lower in central and western regions.

## Econometric model and data

### Model specification

A panel data regression model (1) containing the quadratic term of environmental regulation intensity ($ER^2$) is constructed to evaluate the influence of environmental regulation on regional technological innovation; moreover, in order to examine government financial support'

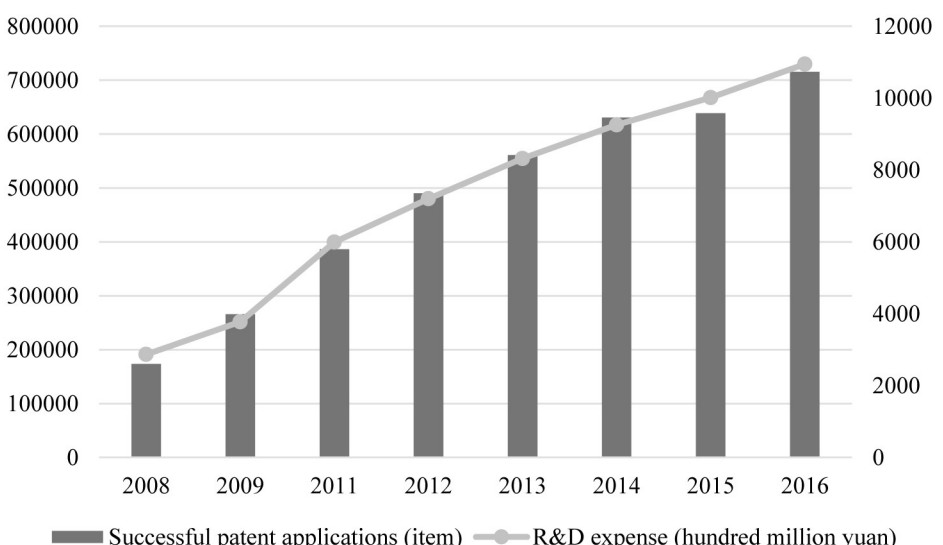

**Fig 4. Corporate R&D expense and successful patent applications from 2008 to 2016 in China.**

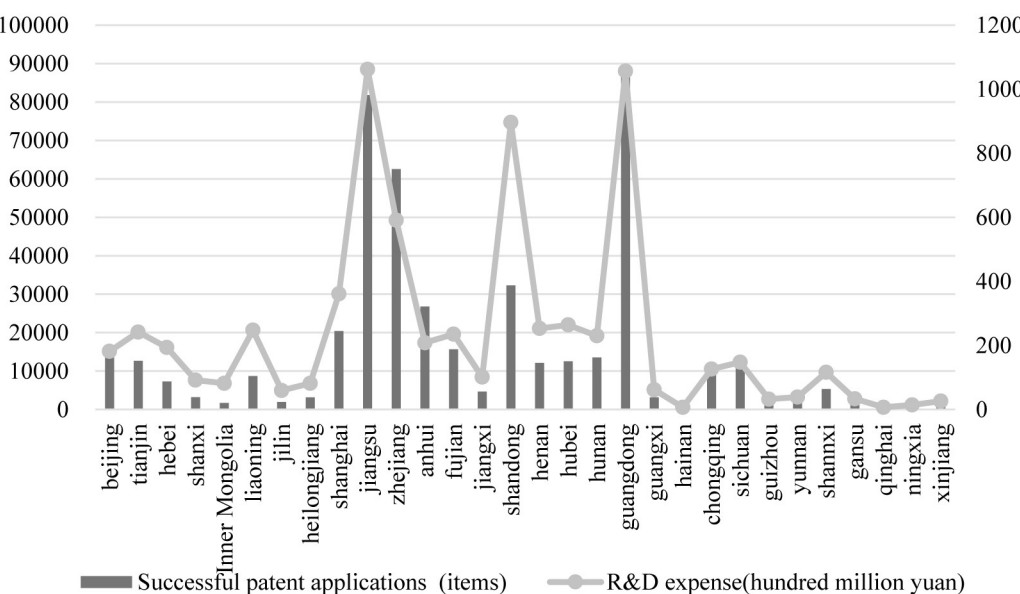

**Fig 5. R&D expense and successful patent applications in Chinese provinces or municipalities.**

indirectly effects on regional technological innovation, the authors incorporate government financial support (*Gov_sup*) into model (1) to construct econometric model (2); further, environmental regulation may be affect, apart from directly working on enterprise technological innovation, corporate technological innovation indirectly through its interaction with government financial support. Thus, the authors integrate the interaction term of environmental regulation intensity and government financial support into model (2) to construct panel data model (3). Specifications of econometric model (1)–(3) are shown below:

$$\ln ETI_{it} = \alpha + \beta_1 \ln ER_{it} + \beta_2 \ln ER^2_{it} + \gamma Control_{it} + \mu_i + \lambda_t + \varepsilon_{it} \tag{1}$$

$$\ln ETI_{it} = \alpha + \beta_1 \ln ER_{it} + \beta_2 \ln ER^2_{it} + \beta_3 \ln Gov\_sup_{it} + \gamma Control_{it} + \mu_i + \lambda_t + \varepsilon_{it} \tag{2}$$

$$\ln ETI_{it} = \alpha + \beta_1 \ln ER_{it} + \beta_2 \ln ER^2_{it} + \beta_3 \ln Gov\_sup_{it} + \beta_4 \ln ER_{it} \times \ln Gov\_sup_{it} + \gamma Control_{it}$$
$$+ \mu_i + \lambda_t + \varepsilon_{it} \tag{3}$$

where *i* and *t* denote region and year respectively. ln*ETI* is the natural logarithm of enterprise technological innovation. ln*ER* signifies the natural logarithm of environmental regulation intensity. ln*Gov_sup* indicates the natural logarithm of government financial support. ln*ER*×ln*Gov_sup* is the natural logarithm of the interaction term of environmental regulation intensity and government financial support. *Control* represent control variables, including human capital (*Hcap*), provincial gross domestic product (*pgdp*), proportion of foreign direct investment in GDP (*fdi*), dependence on foreign trade (*Tra*), and infrastructure (*Infra*). $\mu_i$ and $\lambda_t$ represent entity fixed effects and period fixed effects respectively. $\varepsilon_{it}$ is random error.

## Variables and data

The authors select the industrial enterprises with an annual turnover of over 20 million *yuan* from 30 Chinese provinces or municipalities between 2008 and 2016 as samples to examine the direct and moderating effects of environmental regulation intensity on enterprise

technological innovation. Explained variable of the paper is corporate technological innovation. Environmental regulation intensity and government financial support are explanatory variables. Considering relevant literature and the availability of data, control variables mainly include human capital, provincial gross domestic product, infrastructure, dependence on foreign trade, and proportion of foreign direct investment in GDP. Definitions of variables are as follows.

Explained variables: Enterprise technological innovation (*ETI*). In previous research, various indicators were used to measure corporate technological innovation level, including innovation input indicators like R&D staff, R&D full-time equivalent, R&D expense, and R&D intensity or innovation output indicators such as successful patent applications and sales revenue of new products. In reference to the practice of Greve (2003), and Gemünden (1992), the authors select R&D input intensity (Ratio of R&D expenditure to GDP) in different regions as the indicator to measure enterprise technological innovation [40–41]. High R&D input intensity means high level of corporate technological innovation. Successful patent applications, which indicates corporate technological innovation output, which signifies enterprise technological innovation input are adopted in this paper to conduct robustness test.

Key explanatory variables: Environmental regulation intensity (*ER*). Referring to the practice of Triebswetter & Hitchens (2005), and Zhou et al. (2017), the authors employ paid-in investment in industrial pollution treatment to evaluate environmental regulation intensity [42–43]. In addition, in reference to the ideas of Levinson (1996) and Wang (2002), the authors select pollution control cost per thousand industrial output to represent environmental regulation intensity [44–45]. Of this, pollution control costs are the total costs for the operation of industrial waste water treatment equipment and industrial waste gas treatment equipment, that is,

$$Environmental\ regulation\ intensity = \frac{Pollution\ control\ costs}{total\ industrial\ output} \times 1000 \qquad (4)$$

The larger the value is; the more intense environmental regulation is. This indicator is utilized in the robustness check.

Government financial support (*Gov_sup*). Referring to the practices of Guellec & Van Pottelsberghe de la Potterie (1997) and Guo et al. (2016), the authors adopt the government-financed fund in R&D expense to evaluate government financial support [46–47].

The following control variables are selected.

1. Human capital (*Hcp*). Human capital is a significant driving force for technological innovation. Referring to the measuring methods of Jones (2003) and Yang et al. (2018) [26, 48], the authors utilize average educational attainment to measure the stock of human capital.

2. Provincial gross domestic product (*pgdp*). GDP per capita is employed to measure the development level of different provinces or municipalities. CPI of the region is applied to eliminate the influence of price variance.

3. Economic openness. Referring to the practices of Chen & Feng (2000) and Malesky (2008), the authors select dependence on foreign trade (*Tra*) and proportion of foreign direct investment in GDP (*fdi*) to evaluate economic openness of the region [49–50]. The ratio of total imports and exports to gross domestic product and the ratio of foreign direct investment to gross domestic product are adopted as their respective proxies. The value is calculated in Chinese Yuan (CNY) based on average exchange rate of the US dollar against the CNY.

**Table 1. Descriptive statistics of variables.**

| Variable | Definition | Observation | Mean | Standard Deviation | Minimum | Maximum |
|---|---|---|---|---|---|---|
| ln$ETI$ | Enterprise Technological Innovation (in log) | 270 | 0.940 | 0.520 | 0.100 | 2.197 |
| ln$ER$ | Environmental Regulation (in log) | 270 | 5.141 | 0.891 | 2.510 | 7.256 |
| ln$Gov\_sup$ | Government financial support (in log) | 270 | 10.838 | 1.348 | 5.340 | 12.922 |
| ln$infra$ | Logarithm of Infrastructure | 270 | 2.587 | 0.356 | 1.396 | 3.251 |
| ln$pgdp$ | GDP per capita (in log) | 270 | 10.511 | 0.511 | 9.085 | 11.666 |
| ln$hcap$ | Human Capital (in log) | 270 | 2.239 | 0.111 | 1.945 | 2.594 |
| $Tra$ | Dependence on Foreign Trade | 270 | 0.302 | 0.360 | 0.032 | 1.784 |
| $fdi$ | Proportion of Foreign Direct Investment in GDP | 270 | 0.056 | 0.112 | 0.000 | 0.883 |

4. Infrastructure level. Referring to practices of the previous research, the authors adopt the mileage of road open to traffic to measure infrastructure level.

The data on the above variables is collected from *China Statistical Yearbook 2009–2017*, *China Industry Statistical Yearbook 2009–2017*, *China Statistical Yearbook on Science and Technology 2009–2017*, *China Energy Statistical Yearbook 2009–2017*, and Wind Information Database. Part of indicators are processed in logarithm to realize stationary sequence and eliminate heteroscedasticity. The variables are centralized when analyses of moderating effects are conducted. Table 1 shows the statistical features such as mean value, standard deviation, minimum value, and maximum value. All the variables in monetary terms of this paper was converted into constant prices.

## Results and discussion

### Benchmark regression

This paper uses the panel data of 30 Chinese provinces or municipalities from 2008 to 2016 to explore the direct and moderating effects of environmental regulation intensity on enterprise technological innovation, as well as the effects of government financial support on corporate technological innovation. The results of Hausman test reveal that fixed effects model are better than random effects model. This is because the samples of the research are regions with their own characteristics, which would affect the regression analyses results. To this end, the authors employ fixed effects model, controlling provincial fixed effects and yearly fixed effects, to unravel the direct and moderating effects of environmental regulation intensity on enterprise technological innovation, as well as the effects of government financial support on corporate technological innovation. Regression results are shown in Table 2. From regression results, the authors discover that coefficient estimates of environmental regulation intensity are all negative and significant at the 1% level in model (1)–(3) while coefficient estimates of the quadratic term of environmental regulation intensity are all positive. Which shows that the relationship between environmental regulation intensity and technological innovation of enterprises emerges as a U-shape, and an "inflection point" in the role of environmental regulation in enterprise technological innovation is 5.63 in terms of environmental regulation intensity. Moreover, the threshold value is within the value range of ln$ER$, signifying that China is still at the stage of inhibition before the "inflection point" and environmental regulation intensity should be enhanced.

Government financial support (ln$Gov\_sup$) is integrated into model (2) and the result shows that the coefficient estimates of government financial support is positive, but it isn't significant, demonstrating that government financial support does not significantly work on technological innovation directly. Based on model (2), model (3) is incorporated with the

**Table 2. Benchmark regression results for the whole sample.**

| Variable | Explained Variable: ln$ETI$ | | |
|---|---|---|---|
| | **Model (1)** | **Model (2)** | **Model (3)** |
| ln$ER$ | -0.631*** | -0.632*** | -0.703*** |
| | (-5.85) | (-5.86) | (-6.65) |
| ln$ER^2$ | 0.056*** | 0.056*** | 0.016* |
| | (5.64) | (5.65) | (1.94) |
| ln$Gov\_sub$ | | 0.017 | -0.166*** |
| | | (0.69) | (-3.29) |
| ln$ER$×ln$Gov\_sub$ | | | 0.045*** |
| | | | (4.12) |
| ln$infra$ | 0.244 | 0.240 | -0.198 |
| | (0.51) | (0.45) | (-1.09) |
| ln$pgdp$ | 0.637** | 0.629** | 0.572** |
| | (2.26) | (2.18) | (2.27) |
| ln$hcap$ | 0.350 | 0.356** | 0.432** |
| | (1.28) | (2.30) | (2.53) |
| $Tra$ | -0.016 | -0.018 | -0.003 |
| | (-0.18) | (-0.20) | (-0.03) |
| $fdi$ | 0.061* | 0.068* | 0.006 |
| | (1.77) | (1.85) | (0.07) |
| Constant | 0.829** | 0.717* | 1.955* |
| | (2.38) | (1.66) | (1.80) |
| Time fixed effect | Yes | Yes | Yes |
| Regional fixed effect | Yes | Yes | Yes |
| Observation | 270 | 270 | 270 |
| $R^2$ | 0.574 | 0.573 | 0.601 |

*Notes*: All parameters are estimated based on the fixed effect model; the t statistical value is in parentheses under coefficients;

*, **, and *** represent the significance at the 10%, 5%, and 1% levels, respectively.

interaction term of environmental regulation intensity and government financial support (ln$ER$×ln$Gov\_sup$). The result shows that the coefficient estimate of interaction term is positively significant at 5%. The coefficient estimate of government financial support is significant at 1%, which is insignificant in model (2). This proves that environmental regulation can enhance government financial support's effect on enterprise technological innovation. When

$$\frac{\partial TI}{\partial lnGov\_sub} = \beta_3 + \beta_4 lnER > 0 \tag{5}$$

that is, when the value of ln$ER$ is higher than 3.69, government financial support will facilitate corporate technological innovation. Otherwise, namely, environmental regulation intensity is lower than 3.69, the side effects brought by the interaction between environmental regulation and government financial support overpower the benefits of enterprise technological innovation. Enterprises then would make self-defensive expenditure. Government financial support at such circumstances would be spent to cover discharge fees, pollution fines, or elementary technological innovation like end-of-pipe control, rather than improvements on production techniques and product innovation.

The regression results of control variables reveal that coefficient estimates of provincial GDP in three models are positively significant at 5%, showing local economic growth could

promote enterprise technological innovation. Local economic development lays a solid foundation for the development of enterprises with an annual turnover of over 20 million yuan and boosts technological innovation of these corporates. The regression coefficient of human capital is positive, meaning the improvements on human capital significantly facilitate technological innovation, which keeps in line with extant research. Progress in economic openness, which is indicated by the proportion of foreign direct investment in GDP, would also promote technological innovation in enterprises with an annual turnover of over 20 million yuan as the entry of FDI generates spillover effects on production technologies for local enterprises when an economy cultivates an open environment. This could not only better factor combination in industries of all kind, but adjust the organic composition of technologies in different sectors. In addition, coefficient estimates of infrastructure and dependence on foreign trade are positive, but they aren't significant.

## Robustness check

To ensure the dependability of empirical results, we experiment several robustness checks, such as: replacing explained variables, replacing explanatory variables and utilize one-phase lagged core explanatory variables to estimate the above benchmark regression results.

First, we replace the explained variable with successful patent applications (ln*ETI1*), which reflects corporate technological innovation outputs. Regression results are shown in Table 3.

Moreover, we take pollution control costs per thousand industrial output as the proxy indicator for environmental regulation intensity and mark it as ln*ERs* [38], the authors take pollution control costs per thousand industrial output as the proxy indicator for environmental regulation intensity and mark it as ln*ERs*. Of this, pollution control costs are the total operation costs of treatment facilities for industrial waste water and industrial waste gas. Estimation results are re-calculated and result is displayed in Table 4.

Last, considering the potential endogeneity, the authors replace two key variables in major regression, namely environmental regulation intensity and government financial support, with their lagged variables. The quadratic and interaction term of the above two are also alternated with their lagged terms to mitigate biased error incurred by potential endogeneity. Table 5 shows the result.

The regression results under different robustness checks all indicate that there is no explicit change on the numeric value of coefficient compared with the benchmark regression results, showcasing the consistency of conclusions. The impact of environmental regulation on technological innovation remains a U-shape relation. The sign and significance of the regression coefficient stay unchanged. With particular note, the coefficient of interaction term is also significantly positive, indicating that environmental regulation can enhance government financial support's effect on enterprise technological innovation.

## The heterogeneity across sub-samples

The vast land of China and differences in regional development give birth to regional heterogeneity of the direct and moderating effects of environmental regulation intensity on enterprise technological innovation. The samples are therefore grouped into eastern region samples, central region samples, and western region samples. Of this, eastern region includes *Beijing, Hebei, Shanghai, Zhejiang, Shandong, Tianjin, Liaoning, Jiangsu, Fujian, Guangdong,* and *Hainan*. Central region refers to *Inner Mongolia, Heilongjiang, Jiangxi, Hubei, Shanxi, Jilin, Anhui, Henan*, and *Hunan*. Western region includes *Guangxi, Guizhou, Shaanxi, Qinghai, Tibet, Sichuan, Yunnan, Gansu, Ningxia*, and *Chongqing*. First, a statistical description of the subsamples from three regions is conducted, the results of which is shown in Table 6.

**Table 3. Results of robustness check replacing the explained variable with successful patent applications.**

| Variable | Explained Variable: ln*ETI1* | | |
|---|---|---|---|
| | **Model (4)** | **Model (5)** | **Model (6)** |
| ln*ER* | -0.453*** | -0.445*** | -0.506*** |
| | (-3.32) | (-3.29) | (-3.75) |
| ln*ER*$^2$ | 0.042*** | 0.041*** | 0.006* |
| | (3.35) | (3.30) | (1.83) |
| ln*Gov_sup* | | -0.072 | -0.231*** |
| | | (-0.32) | (-3.59) |
| ln*ER*×ln*Gov_sup* | | | 0.039*** |
| | | | (2.80) |
| ln*infra* | -0.015 | -0.036 | 0.000 |
| | (-0.13) | (-0.29) | (0.00) |
| ln*pgdp* | 0.762* | 0.695* | 0.645** |
| | (1.88) | (1.93) | (2.37) |
| ln*hcap* | 0.584* | 0.561 | 0.627* |
| | (1.69) | (1.63) | (1.85) |
| *Tra* | -0.056 | -0.047 | -0.033 |
| | (-0.50) | (-0.42) | (-0.30) |
| *fdi* | 0.008 | 0.023* | 0.077** |
| | (0.08) | (1.63) | (2.77) |
| Constant | -1.302 | -0.827 | 0.248** |
| | (-0.97) | (-0.61) | (2.58) |
| Time fixed effect | Yes | Yes | Yes |
| Regional fixed effect | Yes | Yes | Yes |
| Observation | 270 | 270 | 270 |
| $R^2$ | 0.335 | 0.348 | 0.367 |

*Notes*: All parameters are estimated based on the fixed effect model; the t statistical value is in parentheses under coefficients;

*, **, and *** represent the significance at the 10%, 5%, and 1% levels, respectively.

Then, regression is conducted on the samples from three regions through fixed effects model. Table 7 shows the estimation results for eastern, central, and western China.

Regression results reveal that estimated results of samples from three regions all showcase a nonlinear U-shape relation between environmental regulation intensity and corporate technological innovation. That is, enterprise technological innovation will fall at first and rise later as environmental regulation continues to be enhanced. The threshold value for reversing the trend is 6.235 in eastern region, 5.567 in central region, and 5.368 in western region. Apparently, the threshold value decreases from eastern to western region, demonstrating that environmental regulation intensity in eastern region to facilitate technological innovation of enterprises with an annual turnover of 20 million yuan is relatively higher while the environmental regulation intensity needed to enable technological innovation of enterprises with an annual turnover of over 20 million yuan in western region is the lowest. This may be attributable to the fact that enterprises in eastern region have a higher productivity, enjoy relatively loose financing environment, and are resourceful in various elements. They are more capable in handling the costs caused by strictly environmental regulation and requires a higher threshold value for environmental regulation to deliver its effects on enterprise technological innovation. Enterprises in central and western region face less favorable environment and their productivity is lower. They are more vulnerable to deal with the costs incurred by

**Table 4. Results of robustness check taking pollution control costs per thousand industrial output as the proxy indicator for environmental regulation.**

| Variables | Explained Variable: *lnETI* | | |
|---|---|---|---|
| | **Model (1)** | **Model (2)** | **Model (3)** |
| ln*ER_s* | -0.207** | -0.210** | -0.396** |
| | (-2.08) | (-2.14) | (-2.32) |
| ln*ER_s*$^2$ | 0.047** | 0.048** | 0.048** |
| | (2.17) | (2.23) | (2.23) |
| ln*Gov_sup* | | 0.076* | 0.030 |
| | | (1.93) | (0.58) |
| ln*ER_s*×ln*Gov_sup* | | | 0.017* |
| | | | (1.96) |
| ln*infra* | -0.063 | -0.021 | -0.036 |
| | (-0.51) | (-0.17) | (-0.29) |
| ln*pgdp* | 1.157*** | 1.065*** | 1.066*** |
| | (4.91) | (4.48) | (4.50) |
| ln*hcap* | -0.743 | -0.627 | -0.694 |
| | (-1.60) | (-1.36) | (-1.50) |
| *Tra* | 0.069 | 0.089 | 0.111 |
| | (0.53) | (0.68) | (0.85) |
| *fdi* | 0.658*** | 0.638*** | 0.652*** |
| | (3.17) | (3.10) | (3.18) |
| Constant | 3.720 | 3.492 | 4.157 |
| | (1.34) | (1.28) | (1.50) |
| Time fixed effect | Yes | Yes | Yes |
| Regional fixed effect | Yes | Yes | Yes |
| Observation | 270 | 270 | 270 |
| $R^2$ | 0.818 | 0.822 | 0.824 |

*Notes*: All parameters are estimated based on the fixed effect model; the t statistical value is in parentheses under coefficients;

\*, \*\*, and \*\*\* represent the significance at the 10%, 5%, and 1% levels, respectively.

environmental regulation. The threshold value for environmental regulation to work on corporate technological innovation is accordingly lower.

Similar to the regression results, coefficient estimates of government financial support are not significant in all regions, meaning the direct effects of government financial support on technological innovation are not obvious in all regions. The comparison between the regression coefficient of the interaction term of environmental regulation and government financial support from different regions in model (3) shows that the regression coefficients of environmental regulation and government financial support in eastern and western region are positive. This proves that environmental regulation can intensify government financial support's effects on technological innovation. Specifically, when ln*ER* in eastern region is higher than 3.542, the enabling effects of government financial support on technological innovation will be in place while ln*ER* in western region has to be higher than 3.792 for government financial support to deliver its effects. In addition, regression results of control variables are basically in line with previous results.

## Conclusions and implications

Currently China faces the struggle between sustaining economic development and inhibiting environment deterioration. One solution to this dilemma is to improve technological

**Table 5. Results of robustness check with lagged explanatory variables.**

| Variables | Explained Variable: ln*ETI* | | |
|---|---|---|---|
| | **Model (1)** | **Model (2)** | **Model (3)** |
| lag_ln*ER* | -0.558*** | -0.558*** | -0.710*** |
| | (-4.79) | (-4.74) | (-5.73) |
| lag_ln*ER*$^2$ | 0.051*** | 0.051*** | 0.021* |
| | (4.74) | (4.69) | (1.81) |
| lag_ln*Gov_sup* | | -0.001 | -0.177*** |
| | | (-0.04) | (-2.99) |
| lag_ln*ER*×lag_ln*Gov_sup* | | | 0.042*** |
| | | | (3.30) |
| ln*infra* | 0.180 | 0.181* | 0.153 |
| | (1.07) | (1.86) | (1.61) |
| ln*pgdp* | 0.768** | 0.769** | 0.626* |
| | (2.50) | (2.49) | (1.73) |
| ln*hcap* | 0.180* | 0.179* | 0.258* |
| | (1.65) | (1.64) | (1.95) |
| *Tra* | 0.129 | 0.129 | 0.134 |
| | (1.37) | (1.37) | (1.46) |
| *fdi* | 0.069** | 0.067** | 0.104* |
| | (2.36) | (2.37) | (1.74) |
| Constant | 0.550** | 0.561** | 1.897* |
| | (2.48) | (2.28) | (1.87) |
| Time fixed effect | Yes | Yes | Yes |
| Regional fixed effect | Yes | Yes | Yes |
| Observation | 240 | 240 | 240 |
| $R^2$ | 0.493 | 0.491 | 0.515 |

*Notes*: All parameters are estimated based on the fixed effect model; the t statistical value is in parentheses under coefficients;

*, **, and *** represent the significance at the 10%, 5%, and 1% levels, respectively.

**Table 6. Descriptive statistics of samples from different regions.**

| | Variable | ln*ETI* | ln*ER* | ln*Gov_sup* | ln*infra* | ln*pgdp* | ln*hcap* | *Tra* | *fdi* |
|---|---|---|---|---|---|---|---|---|---|
| Eastern Region | Mean | 14.745 | 5.481 | 11.391 | 2.590 | 10.914 | 2.313 | 0.627 | 0.036 |
| | Standard deviation | 1.442 | 0.966 | 1.447 | 0.499 | 0.424 | 0.109 | 0.425 | 0.093 |
| | Minimum | 9.618 | 2.542 | 5.340 | 1.396 | 9.751 | 2.113 | 0.097 | 0.000 |
| | Maximum | 16.649 | 7.256 | 12.872 | 3.251 | 11.666 | 2.594 | 1.784 | 0.883 |
| | Observation | 99 | 99 | 99 | 99 | 99 | 99 | 99 | 99 |
| Central Region | Mean | 14.002 | 5.300 | 10.984 | 2.625 | 10.389 | 2.233 | 0.106 | 0.039 |
| | Standard Deviation | 0.691 | 0.590 | 0.845 | 0.213 | 0.364 | 0.062 | 0.042 | 0.035 |
| | Minimum | 12.493 | 3.669 | 8.660 | 2.228 | 9.581 | 2.028 | 0.043 | 0.000 |
| | Maximum | 15.320 | 6.332 | 12.394 | 3.155 | 11.173 | 2.335 | 0.203 | 0.137 |
| | Observation | 81 | 81 | 81 | 81 | 81 | 81 | 81 | 81 |
| Western Region | Mean | 12.839 | 4.624 | 10.098 | 2.550 | 10.176 | 2.164 | 0.120 | 0.083 |
| | Standard Deviation | 1.031 | 0.799 | 1.279 | 0.255 | 0.405 | 0.094 | 0.074 | 0.125 |
| | Minimum | 10.445 | 2.510 | 6.885 | 1.828 | 9.085 | 1.945 | 0.032 | 0.000 |
| | Maximum | 14.771 | 5.972 | 12.922 | 3.142 | 10.959 | 2.361 | 0.411 | 0.641 |
| | Observation | 90 | 90 | 90 | 90 | 90 | 90 | 90 | 90 |

**Table 7. Regression results for eastern, central, and western China.**

| Region | Model | ln*ER* | ln*ER*$^2$ | ln*Gov_sup* | ln*ER*×ln*Gov_sup* | Control Variable | Observation | $R^2$ |
|---|---|---|---|---|---|---|---|---|
| Eastern Region | Model (1) | -0.212* (-1.88) | 0.017* (1.83) | | | Yes | 99 | 0.65 |
| | Model (2) | -0.218* (-1.87) | 0.018** (2.42) | -0.004 (-0.08) | | Yes | 99 | 0.645 |
| | Model (3) | -0.401 (-1.64) | 0.041** (2.50) | -0.255*** (-2.72) | 0.072*** (3.16) | Yes | 99 | 0.685 |
| Central Region | Model (1) | -1.080*** (-5.08) | 0.097*** (4.96) | | | Yes | 81 | 0.844 |
| | Model (2) | -1.126*** (-4.85) | 0.101*** (4.72) | -0.016 (-0.51) | | Yes | 81 | 0.842 |
| | Model (3) | -1.417*** (-4.86) | 0.088*** (3.90) | -0.213* (-1.69) | 0.038 (1.61) | Yes | 81 | 0.846 |
| Western Region | Model (1) | -0.408*** (-2.97) | 0.038** (2.41) | | | Yes | 90 | 0.52 |
| | Model (2) | -0.406*** (-2.93) | 0.037** (2.58) | -0.013 (-0.30) | | Yes | 90 | 0.513 |
| | Model (3) | -0.422*** (-3.08) | 0.014 (0.68) | -0.091* (-1.73) | 0.024* (1.70) | Yes | 90 | 0.527 |

*Notes*: All parameters are estimated based on the fixed effect model; the t statistical value is in parentheses under coefficients;

*, **, and *** represent the significance at the 10%, 5%, and 1% levels, respectively.

innovation intensity. However, technological innovation in China is beset with an array of issues such as homogenous structure, vicious competition, and wasting of resources. Individual entity lacks interior incentive to conduct technological innovation. Vigorous driving force is needed, based on existing innovation policies, to facilitate enterprise technological innovation. Because environmental regulation may be affect, apart from directly working on enterprise technological innovation, corporate technological innovation indirectly through its interaction with government financial support. The paper therefore adopts the industrial enterprises with an annual turnover of over 20 million *yuan* from 30 Chinese provinces or municipalities between 2008 and 2016 as samples to conduct the empirical study. Environmental regulation's role on mediating government financial support's effects on corporate technological innovation is put under limelight. Meanwhile, the authors study whether a regional heterogeneity exists in terms of the effects of environmental regulation and government financial support on enterprise technological innovation.

Our main results can be summarized as follows. First, there exists a U-shaped relation between environmental regulation intensity and technological innovation of enterprises which declines first and then climbs up, and an "inflection point" in the role of environmental regulation in enterprise technological innovation is 5.63 in terms of environmental regulation intensity, and China is still at the stage of inhibition before the "inflection point". Second, the promotion of government financial support to corporate technological innovation is not significant, but environmental regulation can significantly enhance government financial support's effects on enterprise technological innovation. These results remain consistent after we experiment several robustness checks.

Last, region-based samples demonstrate that heterogeneity exists in the influence of environmental regulation intensity and government financial support on corporate technological innovation. Firstly, three regions all showcase a nonlinear U-shape relation between environmental regulation intensity and enterprise technological innovation, but the threshold value for environmental regulation to deliver its effects in eastern region is higher than that of the

central and western region. The authors attributes this to the fact that enterprises in eastern region have a higher productivity and is bestowed with looser financing environment and rich element resources. They are therefore well-positioned to cover the costs caused by strictly environmental regulation. Secondly, the direct effects of government financial support on corporate technological innovation are not obvious in all regions, while environmental regulation can intensify government financial support's effects on enterprise technological innovation. Comparatively speaking, the enabling effects of government subsidy in eastern region on corporate technological innovation are more obvious. In addition, echoing the conclusions of extant research, improvements on control variables such as human capital, GDP per capita, and the proportion of foreign direct investment in GDP significantly promote technological innovation.

Findings in this paper carry broad implications. From the perspective of academic research, incorporating government financial support into the model as an important factor and investigating the direct and moderating effects of environmental regulation intensity on enterprise technological innovation simultaneously provide new insights into develop theories of environmental policy and management. For example, in the establishment and implement of China's the environmental protection policy, which is more important: environmental regulation or government financial support? Whether there is interaction effects between environmental regulations and government financial support policies? How do government departments formulate environmental regulations and financial support policies to improve enterprise innovation? This study provides empirical evidence for better understand—and predict—the relationship of environmental regulation intensity, government financial support, and regional enterprise technological innovation to help guide the research in this filed.

From the perspective of the public policy-making, the research results of this paper show that there exists a U-shaped relation between environmental regulation intensity and technological innovation of enterprises which declines first and then climbs up. But government financial support does not significantly work on technological innovation directly, while environmental regulation drives this effect to be achieved, which highlights the complementarity between environmental regulation and government financial support. It suggests that enacting environmental regulation is the first step, but whether Chinese government can provide an effective incentive mechanism to firms to stimulate their technological innovation practices is another matter [39]. Hence, in order to change this relationship, it is necessary for Chinese government to adopt a hybrid approach combining government financial support and environmental regulation to motivate regional and enterprises' technological innovation. Moreover, the government should further enhance environmental regulation intensity. On the one hand, it could force enterprises to apply technological innovation in pollution control and achieve pollution reduction and control in a high level. On the other hand, it could incentive enterprises to beef up innovation in production techniques and management methods (such as green human resource management), which is benefit for improving their productivity and international competitiveness [51–52]. But the government should also be alarmed that intensive environmental regulation may not be suitable to all region. Instead, the government ought to, based on the actual condition of specific region and sector, implement targeted and differentiated environmental regulation and make timely adjustments, rather than insist on a fixed standard, toward a proper range so as to keep the enabling effects of environmental regulation working [11].

From the perspective of the policy management and operational level, the government should identify the appropriate range for subsidy, but it should be considered synthetically from the whole angle, and adjusted dynamically at any time. Due to the positive externality of innovation and the crowding-out of R&D expense by environmental regulation, lack of

funding in technological innovation is natural. In order to promote the interaction effects of environmental regulation and government financial support on enterprise technological innovation, the government should complement environmental regulation through reasonable funding and policy support. But this does not mean that the government should provide as much subsidy as possible. There exists a threshold value and proper range for environmental regulation to work on technological innovation effectively. When environmental regulation passes the inflection point and begins to enable technological innovation, government subsidy focusing on technological innovation can be cut down or even lifted.

There are some limitations in this paper, which need further consideration. (1) Limitations in the measurement of variables: Existing studies on the relationship between environmental regulation and enterprise technological innovation in China have failed to effectively control for the impact of government financial support because of data limitations. For this research, a government financial support variable, based on the government-financed fund in R&D expense data, is developed and incorporated into the model, and while this variable is relatively consistent with the overall government financial support, there are still cannot distinguish whether it is for environmental-friendly technologies or not. In follow-up studies, additional data on government funding for green technologies need to be compiled, and combined with green innovation for further analysis. (2) Environmental regulations can be further subdivided: For example, it is possible to decompose environmental regulations into different sub-components based on type of regulation, such as the formal and informal environmental regulation. Further, it is also possible that testing the effect of environmental regulations against a wide range of different measures of technological innovation. However, due to manufacturing industry and regional panel data limitations, we are not yet able to examine this issue more extensively. The resulting problems need to be explored and new solutions developed in follow-up research. Therefore, in the latter study, we will try to use micro-enterprise level data to study the interaction effects of environmental regulation and government funding for green technologies on green innovation behavior of enterprises, and consider that in the different types of environmental regulation.

## Author Contributions

**Conceptualization:** Shihu Zhong.

**Data curation:** Fei Song, Junhao Zhu.

**Formal analysis:** Min Deng, Fei Song, Junhao Zhu.

**Funding acquisition:** Shihu Zhong.

**Methodology:** Xiguang Cao, Shihu Zhong, Junhao Zhu.

**Software:** Xiguang Cao, Min Deng, Fei Song, Shihu Zhong.

**Writing – original draft:** Xiguang Cao, Min Deng, Fei Song, Shihu Zhong, Junhao Zhu.

**Writing – review & editing:** Min Deng, Fei Song, Shihu Zhong.

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
