## [Decision Letter · Decision Letter 0]

22 Jul 2019

PONE-D-19-15132

Direct and Moderating Effects of Environmental Regulation Intensity on Enterprise Technological Innovation: The Case of China

PLOS ONE

Dear Song,

Thank you for submitting your manuscript to PLOS ONE. After careful consideration, we feel that it has merit but does not fully meet PLOS ONE’s publication criteria as it currently stands. Therefore, we invite you to submit a revised version of the manuscript that addresses the points raised during the review process.

We would appreciate receiving your revised manuscript by Sep 05 2019 11:59PM. To enhance the reproducibility of your results, we recommend that if applicable you deposit your laboratory protocols in protocols.io, where a protocol can be assigned its own identifier (DOI) such that it can be cited independently in the future. For instructions see: http://journals.plos.org/plosone/s/submission-guidelines#loc-laboratory-protocols

We look forward to receiving your revised manuscript.

Kind regards,

Bing Xue, Ph.D.

Academic Editor

PLOS ONE

Journal Requirements:

We would like to thank the financial support from the Fundamental Research Funds for the Central Universities (CXJJ-2018-361).

Reviewers' comments:

Reviewer's Responses to Questions

**Comments to the Author**

1. Is the manuscript technically sound, and do the data support the conclusions?

Reviewer #1: Partly

Reviewer #2: Yes

Reviewer #3: Yes

2. Has the statistical analysis been performed appropriately and rigorously? 

Reviewer #1: Yes

Reviewer #2: Yes

Reviewer #3: Yes

3. Have the authors made all data underlying the findings in their manuscript fully available?

Reviewer #1: Yes

Reviewer #2: Yes

Reviewer #3: Yes

4. Is the manuscript presented in an intelligible fashion and written in standard English?

Reviewer #1: Yes

Reviewer #2: Yes

Reviewer #3: Yes

5. Review Comments to the Author

Reviewer #1: The paper is interesting and relevant but needs more work.

1. The abstract is too long but not informative.

2. The paper needs language editing.

3. The development of the literature needs more work to link them clearly.

4. The review also needs more recent works.

5. Tables and figures need to be more professionally presented.

6. Table 2 has an R2 of 90+ % is that acceptable or there is some issue of tautology.

7. Discussion needs more work.

8. Implications need to be more specific.

9. There should be some limitations and suggestions for future research.

Reviewer #2: Direct and Moderating Effects of Environmental Regulation Intensity on Enterprise Technological Innovation, it is an important study theme covering different countries and regions all over the world. In this sense, taking the the industrial enterprises with an annual turnover of over 20 million yuan from 30 Chinese provinces or municipalities between 2008 and 2016 in China as an example to explore the Direct and Moderating Effects of Environmental Regulation Intensity on Enterprise Technological Innovation with a panel data regression model does make sense. However, this study did not gain the expected intriguing results or findings and greatly contribute to the previous literature. Thus, I have the following concrete comments:

The literature review has not clearly presented the gap within the current literature, as a result, it is very hard for me to evaluate this paper’s theoretical contribution to the discipline. I do hope the authors could improve this term greatly in the further version.

Totally, the mainly part of the paper brings “little nutrition” to the extension of the research in this field.

In the conclusion, if the author can make a comparative analysis of the research results with previous studies or related studies, it may be helpful for the deepening of this paper.

Reviewer #3: I am afraid that this article is not publishable in PLOS One at current form. There are a few significant issues that require fundamental rethinking.

First and the utmost, the claimed contribution of the paper is not convincing; adding or changing variables do not necessarily imply for added-value to the literature of “Porter hypothesis.” In particular, the indicator of “government financial support” – the government-financed fund in R&D expense – does not make sense since the overall funding can not distinguish whether it is for environmental-friendly technologies or not. When you put this variable in a regression model of technological innovation, which is also not referring to green technologies but the overall innovation efforts, the positive coefficient estimates have very little implication. Take the material industries, for instance, Ministry of Science and Technology released the “13th Five-Year-Plan for S&T Innovation of Materials Industries” in 2017, which proposed to invest substantially on new materials. However, there is only a minimal investment in environmental-friendly materials. Obviously, the model does not work for these industries. The fact is that public R&D investments of green technologies for many other sectors in China are also little. To evaluate the synthetical effect of environmental regulation and public R&D supports on green innovation, singling out public funding for green technologies is necessary, although the data might not be available in all the yearbooks.

Secondly, there are a lot of similar researches published in Chinese peer-review journals in the past five years or so. Many of those articles used the same databases and related variables, however, it’s necessary to have some critical thinkings to push forward the discussion of “Porter hypothesis” in China.

At last, this article is a standard and textbook research in innovation study, which can not be accounted for as interdisciplinary research that PLOS One’s readers are expecting to.

6. PLOS authors have the option to publish the peer review history of their article (what does this mean?). If published, this will include your full peer review and any attached files.

Reviewer #1: Yes: T. Ramayah

Reviewer #2: No

Reviewer #3: No

---

## [Author Response · Author response to Decision Letter 0]

29 Aug 2019

Dear Editor and Reviewers,

Thank you for your letter and for the Reviewers’ comments concerning our manuscript ID PONE-D-19-15132 entitled “Direct and Moderating Effects of Environmental Regulation Intensity on Enterprise Technological Innovation: The Case of China”. These comments were all valuable and very helpful for revising and improving our paper. We have studied the comments carefully and made corrections that we hope will be met with approval. The revised portions are shown in red in our manuscript. The response to the reviewers’ comments are shown in the file “Response to the reviewers”. Moreover, I’m sure that our manuscript meets PLOS ONE’s style requirements, including those for file naming. I have removed the funding-related text from the manuscript. Finally, I stated that “we received no external funding during this specific study, and the funders had no role in study design, data collection and analysis, decision to publish, or preparation of the manuscript.”

The response to the reviewers’ comments are shown in the file “Response to the reviewers”. 

Thank you for your suggestions and comments.

With kind regards,

Sincerely yours,

Shihu Zhong & Fei Song

---

## [Decision Letter · Decision Letter 1]

17 Sep 2019

Direct and Moderating Effects of Environmental Regulation Intensity on Enterprise Technological Innovation: The Case of China

PONE-D-19-15132R1

Dear Dr. ZHONG,

We are pleased to inform you that your manuscript has been judged scientifically suitable for publication and will be formally accepted for publication once it complies with all outstanding technical requirements.

With kind regards,

Bing Xue, Ph.D.

Academic Editor

PLOS ONE

Additional Editor Comments (optional):

Reviewers' comments:

Reviewer's Responses to Questions

**Comments to the Author**

1. If the authors have adequately addressed your comments raised in a previous round of review and you feel that this manuscript is now acceptable for publication, you may indicate that here to bypass the “Comments to the Author” section, enter your conflict of interest statement in the “Confidential to Editor” section, and submit your "Accept" recommendation.

Reviewer #1: All comments have been addressed

2. Is the manuscript technically sound, and do the data support the conclusions?

Reviewer #1: Yes

3. Has the statistical analysis been performed appropriately and rigorously? 

Reviewer #1: Yes

4. Have the authors made all data underlying the findings in their manuscript fully available?

Reviewer #1: Yes

5. Is the manuscript presented in an intelligible fashion and written in standard English?

Reviewer #1: Yes

6. Review Comments to the Author

Reviewer #1: Thank you for carrying out the revisions. Although the literature review is still lacking some clarity overall most of the issues have been covered.

7. PLOS authors have the option to publish the peer review history of their article (what does this mean?). If published, this will include your full peer review and any attached files.

Reviewer #1: Yes: T. Ramayah

---

## [Editor Report · Acceptance letter]

25 Sep 2019

PONE-D-19-15132R1 

Direct and Moderating Effects of Environmental Regulation Intensity on Enterprise Technological Innovation: The Case of China 

Dear Dr. Zhong:

I am pleased to inform you that your manuscript has been deemed suitable for publication in PLOS ONE. Congratulations! Your manuscript is now with our production department. 

With kind regards,

on behalf of

Professor Bing Xue 

Academic Editor

PLOS ONE